# Influence of Proteins on Bioaccessibility of *α*-Tocopherol Encapsulation within High Diacylglycerol-Based Emulsions

**DOI:** 10.3390/foods12132483

**Published:** 2023-06-25

**Authors:** Qian Zou, Weifei Wang, Qingqing Xu, Menglei Yan, Dongming Lan, Yonghua Wang

**Affiliations:** 1School of Food Science and Engineering, South China University of Technology, Guangzhou 510640, China; 18143302986@163.com (Q.Z.); qingqingxu2019@163.com (Q.X.); 13906542522@163.com (M.Y.); yonghw@scut.edu.cn (Y.W.); 2Sericultural & Argi-Food Research Institute, Guangdong Academy of Agricultural Sciences/Key Laboratory of Functional Foods, Ministry of Agriculture and Rural Affairs/Guangdong Key Laboratory of Agricultural Products Processing, No. 133 Yiheng Street, Dongguanzhuang Road, Tianhe District, Guangzhou 510610, China; wangweifei@gdaas.cn; 3Guangdong Yue-Shan Special Nutrition Technology Co., Ltd., Foshan 528000, China

**Keywords:** diacylglycerol, in vitro digestion, bioaccessibility, *α*-tocopherol, delivery systems

## Abstract

*α*-Tocopherol has been widely used in medicine, cosmetics, and food industry as a nutritional supplement and antioxidant. However, *α*-tocopherol showed low bioaccessibility, and there is a widespread *α*-tocopherol deficiency in society today. The preparation of oil-in-water emulsions with high safety and low-calorie property is necessary. The aim of this research was to investigate the effects of different protein emulsifiers (whey protein isolate (WPI), soy protein isolate (SPI), and sodium casein (SC)) on the properties of emulsions delivery system, and diacylglycerol (DAG) was picked as a low-accumulated lipid. The interfacial changes, microstructural alterations, and possible interactions of the protein-stabilized DAG emulsions were investigated during the in vitro digestion. The results show that different proteins affect the degree of digestibility and *α*-tocopherol bioaccessibility of the emulsions. Both WPI- and SPI-coated emulsions showed good digestibility and *α*-tocopherol bioaccessibility (77.64 ± 2.93%). This might be due to the strong hydrolysis resistance of WPI (*β*-lactoglobulin) and the good emulsification ability of SPI. The SC-coated emulsion showed the lowest digestibility and *α*-tocopherol bioaccessibility, this might be due to the emulsification property of hydrolysis products of SC and the potential interaction with calcium ions. This study provides new possibilities for the application of DAG emulsions in delivery systems.

## 1. Introduction

*α*-Tocopherol, one of the most active ingredients in vitamin E (V_E_) family, is composed of a chromogenic ring and a hydrophobic side chain [1]. V_E_ have a variety of health benefits, such as preventing free-radical-mediated oxidative stress-related diseases, reducing diabetes and cancers, and treating neurodegenerative disorders [2,3,4]. However, there is a widespread V_E_ deficiency in society today [5]. A survey showed that in Asia, V_E_ deficiency was present in most age groups, reaching 67%, 80%, 56%, and 72% for adolescents, adults, the elderly, and pregnant women, respectively, which increased the risk of cardiovascular disease (CVD) and liver disease [6]. CVD was even a significant contributor to deaths at present [7]. It was widely suggested that this poor condition was strongly related to low water solubility and low bioaccessibility of V_E_ [8].

It is necessary to develop an efficient and healthy delivery system to improve this situation. Numerous systems have been investigated to deliver lipophilic bioactives, including microemulsions, emulsions, nanoemulsions, multiple emulsions, and multilayer emulsions [9]. Among these, oil-in-water emulsions was considered as a feasible one with an excellent characterization of easy operation [9,10].

Emulsifiers played an essential role in the stability of emulsions and their physicochemical properties. Emulsifiers also greatly impact the rate and extent of lipid digestion and subsequently influence the bioaccessibility of lipophilic bioactives [11]. Among all the emulsifiers, proteins were widely used in the food industry due to their high safety and suitable cost [12,13]. More importantly, it was widely reported that the emulsions stabilized with protein had better lipid digestibility and higher bioaccessibility of active ingredients [14,15]. These reports all suggested that protein was a better emulsifier for delivery systems and was consistent with current consumer health needs [16]. However, protein types showed great influence on bioaccessibility of lipophilic bioactives [13]. Whey protein isolate (WPI) is a byproduct of cheese production. Due to its amphiphilic nature, it was considered to be an excellent natural emulsifier and can be used as a single or co-emulsifier to stabilize emulsions [17]. One of the main components of soybean protein isolate (SPI), 7S (β-conglycinin), has been shown to have good emulsification properties [18]. Sodium caseinate (SC) exhibits excellent emulsification properties due to its hydrophilic and hydrophobic chain segments. However, at acidic pH, the net attraction between casein molecules increases, leading to self-association of adsorbed and non-adsorbed protein components, which may lead to emulsion instability [19]. However, all above-mentioned emulsion systems are lipid-rich systems which are not welcomed due to the prevalence of obesity and related metabolic disorders around the world in the last 40 years [20].

It is urgent to develop a healthier delivery system. Recently, diacylglycerol (DAG) as a low-accumulated fat was widely studied and was considered a good substitute for traditional edible oil [21,22,23]. The current study reported that there seem to be some interactions between DAG and proteins. Even DAG-based emulsions were prepared and studied recently [24,25]; however, the application status of DAG in protein-stabilized emulsions is still unknown. More related studies are still needed to investigate the potential of DAG on delivering bioactive substances.

In order to develop a healthier emulsion delivery system, DAG-based emulsions were prepared. Three different proteins were selected as emulsifiers. The impacts of emulsifiers on the digestion fates of lipid and VE were also investigated. Zeta potential, confocal laser scanning microscopy (CLSM) images, and Fourier-transform infrared spectroscopy (FTIR) were used to investigate the interfacial changes, microstructural alterations, and possible interactions of the protein-stabilized DAG emulsions during in vitro digestion. The digestion of the three emulsions and their ability to deliver V_E_ were investigated by examining acylglycerol profiles and bioaccessibility of V_E_ during in vitro digestion. This study provides important information for new directions in emulsion delivery systems.

## 2. Materials and Methods

### 2.1. Material

DL-α-Tocopherol (96%) and pepsin from porcine gastric mucosa (P110927; ≥3000 units/mg) were purchased from Aladdin Biochemical Technology Co., Ltd. (Shanghai, China). Pancreatic lipase from porcine pancreas (L3126; ≥125 units/mg, using olive oil) was purchased from Sigma-Aldrich Chemical Co., Ltd. (St. Louis, MO, USA). Whey protein isolate (WPI) was obtained from Yuanye Bio-Technology Co., Ltd. (Shanghai, China). Sodium caseinate (SC) and soybean protein isolate (SPI) were purchased from Macklin Biochemical Co., Ltd. (Shanghai, China). Olive-based diacylglycerol (OD) was provided by Guangdong Yue-Shan Special Nutrition Technology Co., Ltd. (Guangdong, China). The acylglycerol profile of OD is shown in Appendix A. All sodium dodecyl sulfate polyacrylamide gel electrophoresis (SDS-PAGE) reagents were obtained from Sangon Biotech Co., Ltd. (Shanghai, China). Ultrapure water was purified by a HetaiMaster Evo-S instrument and used for all experiments. Methanol and *n*-hexane were both HPLC-grade and purchased from Aladdin (Shanghai, China). All other reagents were of analytical grade or superior.

### 2.2. Preparation of Protein-Coated α-Tocopherol Emulsions

*α*-Tocopherol (20 wt% in oil phase) was mixed with OD, and the mixture was stirred at 50 °C for 20 min. WPI, SC, or SPI (0.2 wt% in final emulsion system) were dissolved in ultrapure water and stirred at room temperature for 3 h for totally hydrating the protein. Coarse emulsion was prepared by mixing the aqueous and oil phases in a ratio of 8:2 (*w*/*w*). The final emulsion was obtained after homogenization (3 min, 18,000 rpm) using a high-speed blender (FJ2000-SH, Huxi industrial Co., Ltd., Shanghai, China) [26].

### 2.3. In Vitro Simulated Digestion

Three-stage in vitro digestion model was used to evaluate the digestion behaviors of the prepared emulsions [27]. Due to the absence of starch in the emulsions, we simulated only the dilution of the oral phase. Emulsions (7.5 g) were mixed with simulated saliva fluid (SSF, 7.5 g) and simulated gastric fluid (SGF, 15 g). The pH of the mixture was adjusted to 3.0 using 0.1 M hydrochloric acid. Diluted emulsions were preheated at 37 °C after calcium chloride solution (0.3 M) was added. The mixture was stirred (180 rpm) for 2 h at 37 °C for monitoring the gastric digestion after adding porcine pepsin (2000 U/mL). Subsamples were withdrawn at t = 0, 2, 5, 10, 30, 60 min.

After gastric digestion, bile salt solution (3.75 mL, cholic acid concentration of 100 mg/mL), calcium chloride solution (0.3 M, 60 μL), and the simulated intestinal fluid (SIF, 12.75 g) were added to the gastric chyme, and the pH of the mixture was adjusted to 7.0 using 0.1 M sodium hydroxide. The pancreatic lipase (with final concentration of 2.25 mg/mL) was added to initiate the small intestinal digestion process. The mixture was stirred (180 rpm) for 2 h at 37 °C. Subsamples were withdrawn at t = 5, 10, 15, 30, 60, 120 min, and 4-bromophenylboronic acid was added to stop the reaction.

### 2.4. Bioaccessibility of α-Tocopherol after Digestion

The bioaccessibility of *α*-tocopherol, also known as the micellization rate [28], was determined according to the method described previously [29]. The sample obtained in the end of the small intestinal digestion was centrifuged (11,000 rpm, 50 min, 4 °C). The middle layer (micelle layer) was extracted with a syringe. To extract *α*-tocopherol, 3 mL of hexane/ethanol solution (1/1, *v*/*v*) was added to the digest samples and micelle layer phase. Mixtures were vortexed for 30 s and then centrifuged at 4000 rpm for 2 min. After operation was repeated 3 times, the supernatant was collected and dried under nitrogen to obtain *α*-tocopherol. Finally, the extract was redissolved with methanol and filtered through an 0.45 μm organic filter membrane. The *α*-tocopherol was quantified by reversed-phase high-performance liquid chromatography (HPLC) using the Alliance HPLC System (e2695, Waters, Milford, MA, USA) equipped with a Waters 2489UV/VIS Detector (2489, Waters, Milford, MA, USA). The chromatographic column was a C18 column (250 × 4.6 mm, 5 μm, SunFire). The separation with an isocratic elution (methanol/ultrapure water, 95/5 (*v*/*v*)) was performed using a fluid flow rate of 1.0 mL/min, a column temperature of 30 °C, an injection volume of 10 μL, and the detection wavelength of 295 nm. The regression equation was established from the peak area and standard concentration and R^2^ > 0.99 [29].

Bioaccessibility was calculated by the following equation [13]:$$\mathrm{Bioaccessibility}\;\left(\%\right)=\frac{\alpha \text{-}tocopherol\;concentration\;in\;micelles}{\alpha \text{-}tocopherol\;concentration\;in\;the\;digesta}\times 100\%$$

### 2.5. Determination of Zeta Potential

The ζ-potential test uses electrophoretic light scattering to measure the potential of particles suspended in a sample in a specific solution environment (pH, salinity, additives). The purpose of the test is to examine the charged properties of the particle surface, including electrical properties and potential levels, in order to predict the stability of the entire suspension. The ζ-potential of undigested emulsions and the digested samples was determined using Zetasizer nano series (Malvern Instruments Ltd., Malvern, Worcestershire, UK). The ζ-potential can be calculated using the Henry’ equation as follows:$$U_{E}=\frac{2\varepsilon \gamma f\left(ka\right)}{3\eta }$$
where *U_E_* is the electrophoretic mobility (μm cm V^−1^ s^−1^), *ε* is the dielectric constant, *γ* is the zeta potential (mV), *f*(*ka*) is the Henry’s function, and *η* is the viscosity (cP). The electrophoretic mobility of charged particles was obtained by using laser Doppler velocimetry (LDV). The ζ-potential was calculated from the electrophoretic mobility and applied to the Henry’s equation. The optical parameters were set to the refractive indices of 1.330 for the dispersant (water) and 1.470 for oil. Before the determination, the samples were diluted 100 times with ultrapure water in order to reduce multiple scattering effects [15]. All tests were carried out at room temperature.

### 2.6. Confocal Laser Scanning Microscopy (CLSM)

Confocal electron microscopy (TCS SPE, LEICA, Wetzlar, Germany) was used to examine the microstructural changes of the emulsions during the digestion process. Nile Red solution (50 μL, 0.1% dissolved in ethanol) and FITC solution (50 uL, 0.1% dissolved in ethanol) were mixed with the original emulsion or digested sample (1 mL) to dye the lipids and proteins, respectively [30]. The dyed samples (20 μL) were placed on a slide, and a coverslip was placed on top of the samples. Nile Red and FITC were excited at 532 and 488 nm, respectively [29,30]. All images were taken using a 10 × objective. A minimum of 10 micrographs were recorded for each sample.

### 2.7. Free Fatty Acid Release Analysis

The lipids were extracted using *n*-hexane from the digested samples according to our previous method [26]. The organic phase was centrifuged (4000 rpm, 3 min, 25 °C). The operation was repeated 3 times to collect all the lipids. The acylglycerol profile was analyzed via HPLC according to our previous reported method [26]. The FFA release rate was calculated by the following equation:$$\mathrm{FFA}\;\mathrm{release}\;\mathrm{rate}\;(\%)={\mathrm{FFAs}}_{0}-{\mathrm{FFAs}}_{t}$$
where FFAs_0_ is the initially content of FFAs measured via HPLC, and FFAs_t_ is the content of FFAs measured via HPLC at predetermined points in time during digestion.

### 2.8. Fourier-Transform Infrared Spectroscopy (FTIR)

In order to reduce the interaction between water and other components, the samples were freeze-dried before FTIR testing [31]. The infrared spectra of the samples were obtained on a spectrophotometer (Nicolet IS50, Thermo Electron Co., Waltham, MA, USA) with the range of 4000–400 cm^−1^ at a resolution of 4 cm^−1^.

### 2.9. Sodium Dodecyl Sulphate Polyacrylamide Gel Electrophoresis (SDS-PAGE)

SDS-PAGE analysis was performed on a discontinuous buffer system using 15% separation gel and 5% stacking gel with initial emulsion as a control [13]. Digested sample and the initial emulsion (20 µL) were mixed with 5 µL loading buffer, and the mixture was boiled in boiling water for 5 min. Centrifugation process (10,000 rpm, 1–2 min) was required. A 1 × Tris-Gly electrophoresis buffer (6.04 g Tris, 37.6 g glycine, and 2 g SDS dissolved in water and fixed to 2 L) was used to perform electrophoresis in the electrophoresis tank. A total of 6 µL of supernatant sample was added to the gel, and electrophoresis was started at 80 V. When the samples reached the separation gel, the voltage was adjusted to 120 V. The gels were stained with Coomassie Brilliant Blue R-50 staining solution for 1 h. Later, the gels were decolorized with decolorizing solution for 12 h, and the decolorization solution needed to be changed twice. Photographs were taken with a gel imaging system (QuickGel 6100, Monad, Shanghai, China). TureColor Pre-stained Protein Marker, 2 colors, molecular weights from 10 to 250 kDa, was used in the SDS-PAGE (Tris-Glycine) experiments; the 25 kDa and 70 kDa bands are orange and the rest of the bands are blue.

### 2.10. Statistical Analysis

All test results were performed at least in triplicate, and the results were expressed as mean + standard deviation. Results were subjected to one-way ANOVA and Duncan’s multiple range test using SPSS 27 (SPSS Inc., Chicago, IL, USA) software, and differences were statistically significant when *p* < 0.05.

## 3. Results and Discussion

### 3.1. Changes in Emulsion ζ-Potential during the In Vitro Digestion

The analysis of ζ-potential allowed us to analyze more visually the changes in the interfacial composition during the in vitro digestion. As shown in Figure 1, the ζ-potential values of initial protein-coated emulsions emulsified by WPI, SC, and SPI were −45.07 ± 0.40, −59.47 ± 2.35, and −53.23 ± 1.19 mV, respectively. The isoelectric points (pI) of these proteins were all between 4 and 5 [32]. The ionization of amino acids, leading to the presence of amino (-NH_2_) and carboxyl (-COOH) groups, governs the surface charges of droplets in protein-coated emulsions, and their behavior is influenced by the pH level of the protein-coated emulsions. When the pH was higher than the pI, the carboxyl group became negatively charged (-COO^−^), while the amino group remained neutral (-NH_2_), resulting in a negatively charged emulsion [33]. This result was in agreement with Liu et al. (2021) [24]. The ζ-potential values of three protein-coated emulsions were significantly different, which could be caused by structural differences between the different proteins and the interactions between proteins and DAG [34]. Due to the competition for the interface between DAG and proteins, the protein adsorption amounts on the oil-lipid surface of TAG emulsions were high in comparison to those of DAG emulsions, resulting in lower ζ-potential values in the later system [35].

The absolute value of ζ-potential for all gastric digesta decreased. In the end of the gastric phase, the ζ-potential values of WPI-, SPI-, and SC-coated emulsions were −41.97 ± 3.25 mV, −42.77 ± 1.23 mV, and −30.97 ± 1.00 mV, respectively. The pH of the gastric phase was 3.0 which was obviously lower than the pI of all mentioned proteins. For these three proteins, the ζ-potential values of the digestion systems should be negative, which is inconsistent with the results of this paper. It could be explained by the fact that after 30 min of the gastric digestion, part of the surface proteins was hydrolyzed [36], which led to lower electrostatic repulsion between oil droplets. This phenomenon was also reported by Chen, Yokoyama, et al. (2020) [37]. The trend of ζ-potential exhibited by DAG emulsions during the gastric digestion was similar to that observed in the TAG emulsions reported by Liu et al. (2021) [24]. However, the absolute value of ζ-potential was significantly higher for DAG emulsions. The different states of the protein adsorbed layer on the surface of TAG and DAG oil droplets should be responsible for this phenomenon [35].

All emulsions showed a notable reduction in ζ-potential values after the intestinal digestion, reaching levels as low as −90 to −100 mV. The first reason is that the protein charge at pH 7 is much lower than at pH 3. This also could be attributed to the replacement of little undigested protein by bile salts at the interface [38]. Furthermore, no significant difference was found in ζ-potential values in the end of the intestinal digestion for these three protein-coated emulsions (*p* > 0.05, −95.47 ± 1.18 mV, −89.93 ± 5.36 mV, and −97.23 ± 2.87 mV for WPI-, SPI-, and SC-coated emulsions, respectively). Interestingly, the ζ-potential values of the DAG-based emulsions in this study were significantly higher than those of the TAG-based emulsions as previously reported (around −80 mV) [39]. It is known that DAG has a higher hydrolysis rate than TAG. More negatively charged FFAs were produced and might assemble at the interface, which could be the main reason for the above phenomenon [24,40].

### 3.2. Microstructure Changes during the In Vitro Digestion

The microstructure images of the three protein-coated emulsions during digestion were measured by CLSM and are presented in Figure 2. Lipid was dyed red, and protein was presented in green. As shown in Figure 2a,d,g, the initial SC-based emulsions displayed uniform droplet distribution without notable aggregation or flocculation. However, the WPI- and SPI-coated emulsions exhibited considerable protein coagulation, likely due to lower exposed hydrophobicity of the protein [41]. Similar results were reported by Lin et al. (2021) [42], who explored the microstructure of WPI- and SPI-coated TAG emulsions. During the gastric digestion, the protein-coated emulsions were easily flocculated due to the weakening of the electrostatic shielding effect by the hydrolysis of the proteins [11,43]. However, no significant aggregation was observed for the SC- and SPI-coated emulsions, the high absolute value of the ζ-potential could explain this phenomenon [44]. This result was also confirmed by Iddir et al. (2020) [45], who investigated the microstructure changes during the in vitro digestion in SC- and SPI-coated TAG emulsions. Similarly, aggregation was not observed for the WPI-coated emulsions. The hydrolysis product of WPI (*β*-lg) was resistant to pepsin digestion, and *β*-lg also showed good emulsification properties [46]. 

After the intestinal digestion, aggregation was still not observed in the WPI-coated emulsion, this might also be caused by the hydrolysis product of WPI (*β*-lg). Moreover, the SPI-based systems exhibited no significant aggregation. This might be attributed to the fact that the SPI hydrolysate had stabilizing properties for emulsions [47]. However, in the SC-coated emulsion, significant aggregation was observed. It might also be due to changes in oil droplets’ surface composition as the interfacial proteins of the oil droplets were further hydrolyzed into small molecules of peptides. The interface of SC-coated emulsions was occupied by the bile salts and hydrolysis products (mainly peptides and FFAs), leading to a lower viscosity and elasticity and making the droplets more prone to aggregation [48].

### 3.3. Fourier Transform Infrared Spectroscopy Analysis

To explore the interaction between particles during the in vitro digestion of protein-coated emulsions, the changes of chemical structure were investigated by FTIR spectroscopy (Figure 3 and Appendix A). In the three initial freeze-dried samples, two peaks appeared around 2930 cm^−1^ and 2860 cm^−1^, which was attributed to symmetric and asymmetric stretching vibrations of C-H in the alkyl skeleton of the OD [49,50]. The absorption peak appeared around 1740 cm^−1^ due to the C=O stretching vibration in OD molecule, which was similar to the characteristic peak of olive triglycerides [51]. The peaks of *α*-tocopherol at 1462 cm^−1^ and 1377 cm^−1^ were observed in the three initial freeze-dried samples, which were generated by shear vibrations of the methylene and methyl groups on its side chains [52,53]. In addition, in the three initial freeze-dried samples, peaks were in the region of 3200 cm^−1^ to 3600 cm^−1^; this was related to the stretching vibration of O-H [54]. There was a difference in this absorption peak between different proteins. This might be due to the strength of the intermolecular hydrogen bonds formed between different proteins and DAG [55]. The absorption peaks were weakened and shifted during the in vitro digestion; this might be due to the disruption of the hydrogen bonding network. The amide I band of the protein (1700 cm^−1^–1600 cm^−1^) was sensitive to changes in the secondary structure [56]. For SPI freeze-dried samples, the absorption peak at 1629.5 cm^−1^ was located in the amide I band, which was caused by the stretching vibration of C=O in the β-folded structure. It shifted to 1652.7 cm^−1^ in the end of the gastric digestion, probably due to irregular coating [57]. In the end of the intestinal digestion, the strength of this absorption peak almost disappeared, which was attributed to changes in the protein structure. For both WPI and SC freeze-dried samples, two peaks appeared corresponding to the amide I band (C=O stretching vibration) and the amide II band (C-N stretching vibration and N-H bending vibration) of the protein [58], respectively. However, in the end of the gastric digestion, these two absorption peaks underwent different degrees of weakening and shifting, which might be due to the hydrolysis of the amide bond of the protein. These two absorption peaks were not detected in the end of the intestinal digestion.

### 3.4. Proteolytic Hydrolysis during SGF Incubation

The digestion of proteins of three protein-coated emulsions was characterized via SDS-PAGE and is shown in Figure 4. A certain amount of trailing was observed for all mentioned emulsions, which was probably due to the high concentration of salt ions in the digestive system [59]. In the initial WPI-coated emulsions, the proteins were mainly concentrated around 14–20 kDa and 66 kDa. The bands of 14–20 kDa were mainly related to the *β*-lactoglobulin (*β*-LG) (18.2 kDa) and α-lactalbumin (*α*-LA) (11.4 kDa). After the gastric digestion, the bands at 66 kDa and 11.4 kDa (mainly *α*-LA)) disappeared, while the majority of bands at 18.2 KDa (mainly *β*-LG) remained intact, which was consistent with previous studies which reported that *β*-LG was more resistant to protein hydrolysis [13,39]. Simultaneously, new low-molecular-weight peptides (<10 kDa) were observed. In the initial SPI-coated emulsion, the major subunits of soy globulin and β-accompanying soy globulin could be clearly observed [60]. Moreover, only low-molecular-weight- peptides could be observed after the gastric digestion. SPI showed no resistance to pepsin and caused this result [61]. In the initial SC-coated emulsion, the bands were mainly concentrated in 25–35 kDa, which was the main component of casein (*β*- and *α*_s_-casein) [62]. The bands above 40 kDa were mainly some protein aggregates [59]. In the end of the gastric digestion, the oil droplets’ surface of SC-coated emulsions was coated with small-molecule peptides (below 10 kDa), which was similar to the SPI-coated emulsion.

### 3.5. Analysis of FFA Release

To analyze the influence of protein emulsifiers on the digestion fates of OD, the FFA release amount and the FFA release first-order kinetics were analyzed, and the results are shown in Figure 5 and Appendix A. The release of FFAs for all emulsions was rapid in the first 15 min, and then the release rate gradually decreased [63]. This might be related to the decrease of substrate amounts [15], while the accumulation of lipolytic products at the oil-water interface could lead to the decrease of contact area between pancreatic lipase and lipids [64].

Except for the similar trend, these three emulsions also showed significantly different FFA release rates and amounts. The WPI-coated emulsion exhibited the fastest FFA release rate followed by SC-coated emulsion and SPI-coated emulsion. This might be due to the removal of the lipolysis products from the surface of the oil droplets by the hydrolysis products of WPI (*β*-lactoglobulin) and higher resistance to displacement of bile salts [65]. At the same time, the SPI has the capability to retard lipolysis of emulsions [66]. The presence of bile salts has been reported to influence the rate and extent of lipid digestion, and the bile salts can act as a cofactor for pancreatic lipase, resulting in higher lipase activity [67]. The low ζ-potential of SC-coated emulsion also confirmed these results.

In the end of the intestinal digestion, the SPI-coated emulsion and WPI-coated emulsion showed a similar FFA release amount (52.39 ± 0.38% and 51.58 ± 0.80%, respectively) and lipolysis degree (60.94 ± 0.35% and 60.48 ± 0.38%, respectively), followed by the SC-coated emulsion (49.32 ± 2.36%); this was in agreement with previous studies [13]. The low emulsification property of hydrolysis products of SC might have contributed to this phenomenon, thus leading to the lower contact area between the lipids and pancreatic lipase [37]. SC also had the potential to bind calcium ions, and their interaction would reduce the amount of calcium ions present in the system [68]. Calcium ions have a facilitating effect on the digestion of OD [69].

### 3.6. Bioaccessibility of α-Tocopherol

The bioaccessibility of *α*-tocopherol, also known as the micellization rate, was investigated and is shown in Figure 6. The lipolysis products of lipids formed mixed micelles with the bile salts and other substances, which were necessary steps for the absorption of lipid-soluble components [70]. Different protein-coated emulsions showed different bioaccessibility. The highest bioaccessibility of *α*-tocopherol was found for SPI (77.64 ± 2.93%), followed by WPI (66.07 ± 7.61%) and SC (63.72 ± 7.22%). The bioaccessibility was positively correlated with the amount of free fatty acids, in agreement with other studies [71,72]. The lowest FFA release amount of SC-coated emulsions was in accordance with the lowest VE bioaccessibility. In addition to this, the highest level of MAG of SPI-coated emulsions after the intestinal digestion process (Table 1) could enhance the formation of micelles and lead to higher bioaccessibility of *α*-tocopherol.

## 4. Conclusions

In conclusion, this study indicated that DAG showed good potential in the delivery system of lipophilic active ingredients based on an in vitro digestion model. The DAG-based system not only showed lower accumulation property but also good *α*-tocopherol bioaccessibility. Meanwhile, different protein emulsifiers (WPI, SC, SPI) showed great impacts on the formation of this delivery system. The SPI-coated emulsions showed the highest emulsion stability and *α*-tocopherol bioaccessibility. This might be due to the best emulsification property of SPI, which improves the contact between pancreatic lipase and lipids and later improves the formation of micelle. This study provides another plausible direction for a healthier delivery system.

## Figures and Tables

**Figure 1 foods-12-02483-f001:**
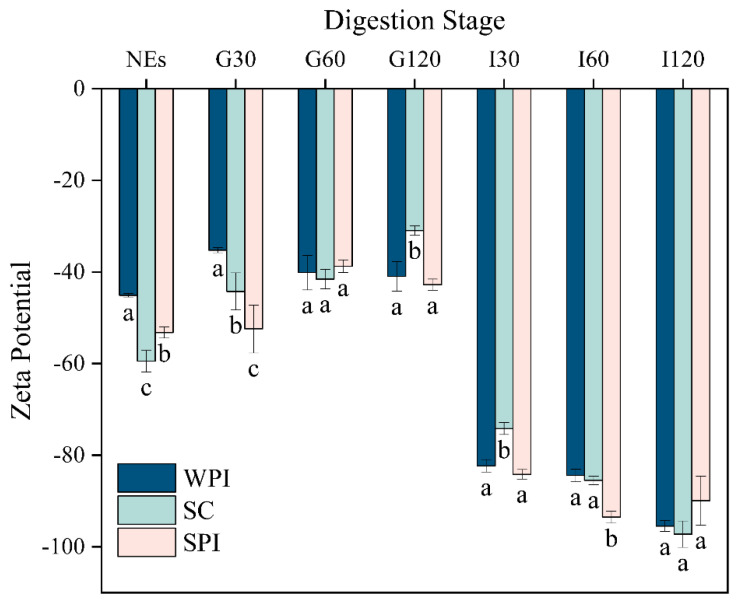
Values of ζ-potential (mV) of WPI-, SC-, and SPI- coated emulsions during the in vitro digestion. NEs means initial emulsions, G means during gastric stage, I means during intestinal stage, and the number means the time of each stage of incubation (G30 means gastric digestion of 30 min). Different letters indicate significant differences between the different groups, *p* < 0.05.

**Figure 2 foods-12-02483-f002:**
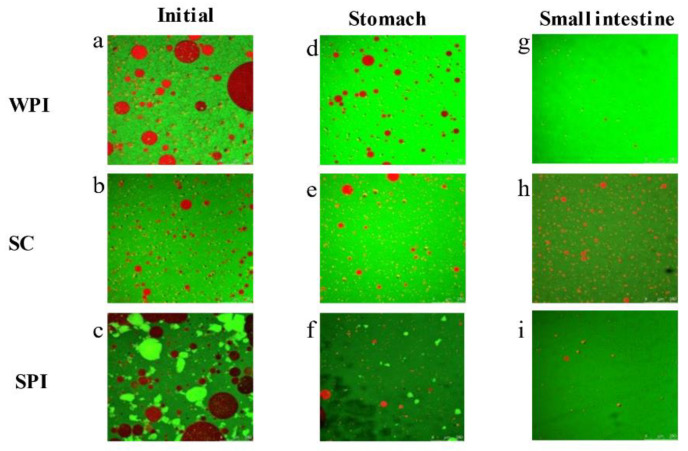
Microstructure of emulsions coated by different types of protein emulsifiers in different simulated GIT stages.

**Figure 3 foods-12-02483-f003:**
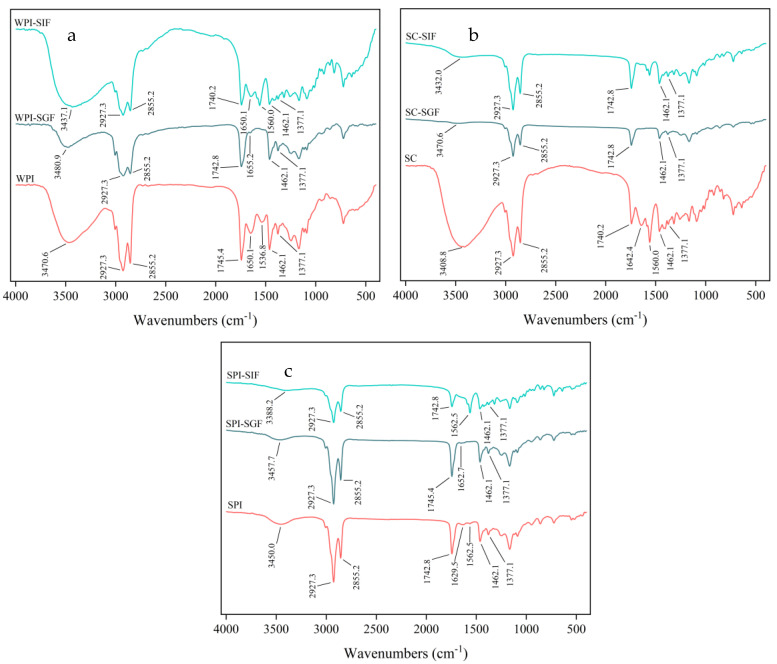
FTIR spectra of WPI-based (**a**), SC-based (**b**), and SPI-based (**c**) emulsions at different stages of digestion.

**Figure 4 foods-12-02483-f004:**
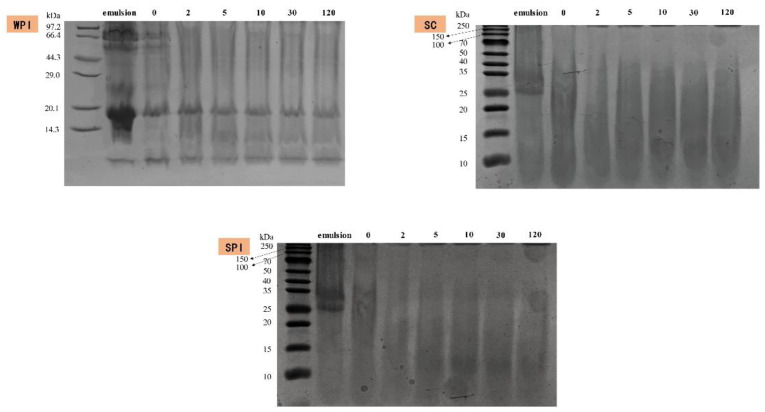
SDS-PAGE patterns of WPI-, SC- and SPI-coated emulsions during incubation with simulated gastric fluid (SGF). The number above the electropherogram means the time of each stage of incubation.

**Figure 5 foods-12-02483-f005:**
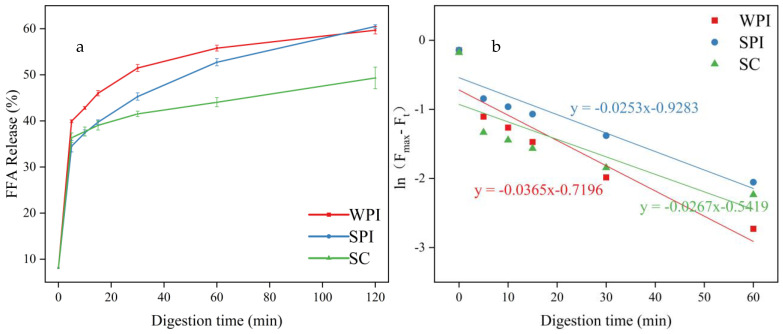
(**a**) Free fatty acids (FFAs) released from emulsions coated by different proteins during the in vitro intestinal digestion. (**b**) Plots of the first-order kinetics of FFA release as a function of digestion time. (The slope indicates the apparent rate constants k (s^−1^) in the fitting equation.).

**Figure 6 foods-12-02483-f006:**
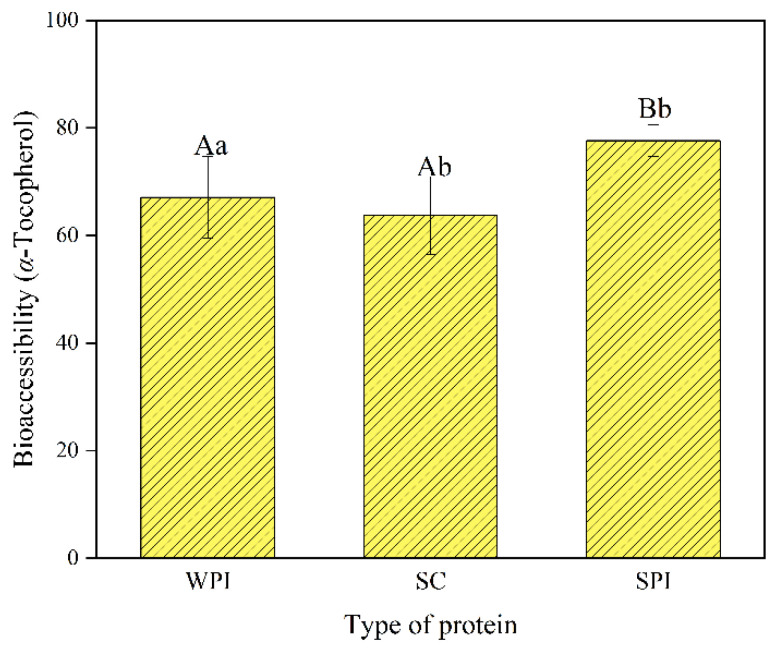
Impact of protein type on the bioaccessibility of *α*-tocopherol determined by measuring the fraction solubilized in the micelle phase after digestion. Samples denoted with upper case letters (A, B) and lowercase letters (a, b) were significantly different (*p* < 0.05).

**Table 1 foods-12-02483-t001:** Acylglycerol profiles of WPI-, SC-, and SPI-coated emulsions at the end of intestinal digestion.

	TAG	FFAs	1,3-DAG	1,2-DAG	1-MAG	2-MAG
WPI	7.21 ± 0.72 ^a^	59.67 ± 0.80 ^a^	12.99 ± 0.56 ^a^	6.12 ± 0.60 ^a^	9.55 ± 0.24 ^a^	4.44 ± 0.04 ^a^
SC	10.02 ± 0.55 ^b^	49.32 ± 2.36 ^b^	23.10 ± 2.14 ^b^	9.59 ± 0.93 ^b^	4.24 ± 0.64 ^b^	3.75 ± 1.06 ^a^
SPI	6.55 ± 1.19 ^a^	60.48 ± 0.38 ^a^	11.06 ± 1.14 ^a^	5.39 ± 0.60 ^a^	9.50 ± 0.22 ^a^	7.02 ± 0.15 ^b^

Different letters indicate significant differences between the different groups, *p* < 0.05.

## Data Availability

The data presented in this study are available on request from the corresponding author.

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
