# Peer review of "Influence of Proteins on Bioaccessibility of α-Tocopherol Encapsulation within High Diacylglycerol-Based Emulsions"

_foods, 2023, doi:10.3390/foods12132483_

Round 1
Reviewer 1 Report
Overall, the content of manuscript is good only minor revisions are needed.
Please make sure figure can be understood without needed to look elsewhere in manuscript. Explain abbreviations in figure legends. E.g. Figure 2 can only be understood, if consulting the method description.
Figure 4 could be bigger and clearer.
Please, check formation throughout; spaces are missing or are too much, font size and alignment are wrong, etc…
Reviewer 2 Report
The research project is very interesting. The use of alpha-tocopherol in the food industry is a powerful antioxidant that can be used naturally. Studies show that at the in-vitro level the results are favorable. However, a protocol for the best vehicle in a food must be established in order to complete this objective. An in vivo study must even be carried out to corroborate the antioxidant effect.
Author Response
请参阅附件。

Reviewer 3 Report
The article titled " Influence of proteins on Bioaccessibility of α-Tocopherol Encapsulation within high diacylglycerol-based Emulsions" aims the effects of whey protein isolate (WPI), soy protein isolate (SPI) and sodium casein (SC)) on the properties of emulsion delivery system. The manuscript presents interesting results, in order to improve it, I made the following observations:
1. Separate the following text in a paragraph:
Emulsifiers played an essential role in the stability of emulsions and 47.their physicochemical properties. Emulsifiers also greatly impact the rate and extent of 48
lipid digestion, and subsequently influenced the bioaccessibility of lipophilic bioactives 49
(Zhang et al., 2015a). Among all the emulsifiers, proteins were widely used and focused 50
in the food industry due to their high safety and suitable cost (Iddir et al., 2022; L. Chen, 51
Yokoyama, et al., 2020). More importantly, it was widely reported that the emulsions sta- 52
bilized with protein had better lipid digestibility and higher bioaccessibility of active in- 53
gredients (Park et al., 2018; Lv et al., 2019). These reports all suggested that protein was a 54
better emulsifier for delivery systems and was consistent with current consumer health 55
needs (Bai et al., 2016). However, protein types showed great influence on bioaccessibility 56
of lipophilic bioactives (Chen et al., 2020). However, all above-mentioned emulsion sys- 57
tems were lipid-rich systems, which were not welcomed due to the prevalence of obesity 58
and related metabolic disorders around the world in the last 40 years (Iacobini et al., 2019).
2. It is necessary to describe the emulsifying properties of soy protein and sodium caseinate in the introduction.
3. Section 2.5. Determination of zeta potential. It is required to describe the technique, insert the equations used to determine the zeta potential
4. Section 2.7. Free fatty acid release analysis. Describe the characteristics of the centrifuge.
5. Describe in the legend of Figure 1, what means NEs, G30, G40...
6. LINE 193. What was the pH of those emulsions?
7. This is not clear "The pH of the gastric phase
was 3.0 which was obviously lower than the pI of all mentioned proteins, the ζ-potential
values of the digestion systems should be negative which was inconsistent with the results
of this paper." By example: the lower de pH, the higher the zeta potential. The pI of protein 1LPK (https://www.rcsb.org/structure/1IPK) is pH 6.16, at pH 3 its charge is 139 e (using PROPKA).
8. LINE 226. The protein charge at pH 7 is much lower than at pH 3. That is why a zeta potential of -100 mV is observed. The structure and function of most macromolecules are influenced by pH.
9. The following sentence is not correct "It was attributed to the replacement of
proteins by bile salts at the interface (Xiang et al., 2023)", a salt cannot replace the large volume of proteins.
10. The following sentence is not confirmed, delete it from the manuscript: This could be attributed to the fact that
bile salts was the primary component of the oil-water surface for all these three protein-
coated emulsions in the end of intestinal digestion (Sandra et al., 2008).
11. Indicate in the images of figure 3, the different states of digestion: Figure 1a, Figure 1b, Figure 1c.
12. What do the letters A and B in Figure 6 mean?
Author Response
请参阅附件。

Round 2
Reviewer 3 Report
1. Insert a superscript in the legend of table 1 to indicate the footnote.